# TENT: Efficient Quantization of Neural Networks on the tiny Edge with Tapered FixEd PoiNT

## ABSTRACT

In this research, we propose a new low-precision framework, TENT, to leverage the benefits of a tapered fixed-point numerical format in TinyML models. We introduce a tapered fixed-point quantization algorithm that matches the numerical format's dynamic range and distribution to that of the deep neural network model's parameter distribution at each layer. An accelerator architecture for the tapered fixed-point with TENT framework is proposed. Results show that the accuracy on classification tasks improves up to ≈31% with an energy overhead of ≈17-30% as compared to fixed-point, for ConvNet and ResNet-18 models.

## KEYWORDS

deep neural networks, low-precision arithmetic, tapered fixed-point

## 1 INTRODUCTION

In the last decade, there has been a surge in deep neural network (DNN) development and deployment for a wide range of use-cases, from bio-medicine [34] to precision agriculture [27]. One of the reasons for this success can be attributed to the dramatic enhancement in the knowledge capacity of DNN models. For instance, the knowledge capacity of DNNs for language translation has boosted by 629x from the GNMT model (278 million parameters) [38] to the recent GPT-3 model (175 billion parameters) [4] within a four year timespan. Increasing the knowledge capacity of DNN models also increases the number of operations, mostly multiply-and-accumulate (MAC), at trillions of operations per ML inference [15]. However, these large networks, classified as CloudML models are deployed on cloud based datacenters where each node utilizes massive compute resources which require hundreds of watts of power(*eg.*, RTX-3070 Nvidia GPU has 8 GB memory, with 20 TFLOPs throughput, and 220 watt power consumption).

Tangential to the trend of upscaling cloudML models to improve performance, a new class of models have emerged. MobileML [33, 35] and TinyML [2, 3, 12, 22, 23, 30] models address the rapidly growing demand to deploy DNNs on the *edge* and on the *tiny edge* (<1W) devices.

To deploy these models on such resource constrained *tiny edge* platforms (*eg.*, ARM M-7 MCU with 2 MB, 216 million cycles per second (MCPS) throughput, and 0.3 watt power consumption [2]), the TinyML models are either designed from scratch through neural architecture search (NAS) [2, 22, 23] or by compressing a version of the CloudML models. Often these compression mechanisms include approaches such as low-precision arithmetic [9, 21, 30], model pruning [12], knowledge distillation [5], and low-rank approximation [36]. Of the aforementioned techniques for TinyML models, low-precision arithmetic has been gaining significant traction. With this technique, the performance of the models can be compromised based on the numerical format selected.

TinyML models with low-bit precision have been deployed on *tiny edge* devices [30]. Unfortunately, reducing the bit-precision in fixed-point numerical formats (binary/ternary in extreme cases) can jeopardize the model performance due to the limited and fixed dynamic range [26, 29]. Equispaced distribution of values expressed by these numerical formats exacerbates the issue to some degree [26]. Pre- and post-processing approaches such as quantization aware training (QAT) [7, 16, 18], retraining [24] and calibration [25] to boost the performance of TinyML models using fixed-point increase their computational complexity. To overcome the performance loss with minimal hardware overhead, a new numerical format with unequal-magnitude spacing (tapered accuracy) and flexible dynamic range is needed. The tapered fixed-point format [20] offers both of these characteristics lacking in the conventional fixed-point format.

In this introductory paper, we are motivated to evaluate the efficacy of the tapered fixed-point numerical format compared to the standard fixed-point numerical format on multiple benchmarks. To study the performance of this new numerical format, a new framework, TENT, has been developed that quantizes TinyML model parameters to tapered fixed-point. The dynamic range and distribution (tapered or uniform) of the numerical format is adapted proportional to the dynamic range and distribution of parameters of each layer in the model. Furthermore, a hardware architecture is designed to study the complexity of the approach compared to fixed-point for ≤ 8-bit TinyML model inference in terms of latency and energy (on the CIFAR-10 dataset).

The key contributions of this work are as follows:

(1) a tapered fixed-point quantization algorithm that adapts the numerical format to best represent the layerwise dynamic range and distribution of parameters within a TinyML model.
(2) a low-precision deep learning framework, TENT, that demonstrates better performance of tapered fixed-point over fixed-point formats for multiple classification tasks.
(3) an implementation of the TENT framework as a custom hardware architecture to study the latency and energy consumption of tapered fixed-point vs standard fixed-point.

## 2 TAPERED FIXED-POINT FORMAT

The tapered fixed-point numerical format (TFX) [20] can be illustrated as a combination of the posit [13] and fixed-point numerical formats. It combines the hardware-oriented characteristics of fixed-point and the high accuracy of posit with tapered precision; where the values are distributed in a non-uniform tent shape which closely resemble the shape of DNN statistics. Specifically, the binary encoding to represent the integer bits in fixed-point is replaced with the signed unary encoding that was previously used to represent the regime bits in posit [13]. This change adds tapered precision characteristics to the fixed-point numerical format. The fraction remains the same as the standard fixed-point where the fraction is added to the integer rather than scaling, which reduces

the hardware complexity as compared to posit and floating-point. The tapered fixed-point is defined as TFX($n$,$IS$) where $n$ refers to the total number of bits and $IS$ (in a range of $[1, n]$) indicates the maximum number of unary encoded integer bits. The $IS$ value controls both the dynamic range as in (1) and the tapered precision. The dynamic range and tapered precision are scaled proportional to $IS$. For instance, the TFX($n$, $IS$) numerical format with $n = 5$, $IS = 1$ and $IS = 2$ has the two smallest dynamic ranges $(\frac{1}{0.0625})$ and $(\frac{2}{0.125})$ respectively, and behaves similar to fixed-point numerical format with uniform precision. Also, $n = IS = 5$ represents the maximum dynamic range $(\frac{5}{0.125})$ and maximum tapered precision.

$$D_{TFX} = \begin{cases} \frac{1}{2^{n-1}}, & \text{if } IS = 1 \\ \frac{Is}{2^{n-2}}, & \text{if } otherwise \end{cases} \quad (1)$$

The value of a TFX number is represented by (2), where $I$ is computed in (3) representing the integer value, $f$ indicates the fraction value and $fs$ the maximum number of bits allocated for the fraction. Note that, the TFX number can also be scaled up/down by 2 raised to the power $SC$.

$$x = I + \frac{f}{2^{fs}} \quad (2)$$

The integer bit-field is encoded based on the *runlength m* of identical bits ($I...I$) terminated by either an *integer terminating bit* $\bar{I}$ or the end of the $n$-bit value. Note that the sign bit is flipped and is also considered as the first bit of the integer.

$$I = \begin{cases} -m, & \text{if } I = 0 \\ m - 1, & \text{if } I = 1 \end{cases} \quad (3)$$

For instance, 3.875 in the TFX(8,8) numerical format is represented by 3 as an integer ($I$), 0.875 as a fraction ($f$), as shown by Fig 1. More details about the tapered fixed-point number format can be found in [20].

$$X_{TFX} = \boxed{S/I \mid I \mid I \mid I \mid \bar{I} \mid f_2 \mid f_1 \mid f_0}$$

$$X_b = \boxed{0/1 \mid 1 \mid 1 \mid 1 \mid 0 \mid 1 \mid 1 \mid 1}$$

$$X_d = 3 + 0.875 = 3.875$$

**Figure 1: Representation of a number in the tapered fixed point TFX(8,8) format**

## 3 RELATED WORK

Studies considering low-precision arithmetic have experimentally shown that TinyML models using 8-bit fixed-point numbers can achieve inference accuracy comparable to that of 32-bit floating-point numbers [2, 9, 21–23]. However, it requires either pre-processing such as calibration [25], quantization aware training (QAT) [7, 16, 18], or post-processing such as retraining [24]. For instance, Banbary *et al.* demonstrate the efficacy of 8-bit fixed-point for TinyML models on the visual wake words (VWW) dataset [2]. The outcome of this study, which uses the QAT approach to quantize weights and activations, indicates that 8-bit fixed-point model parameters

are sufficient to achieve inference performance comparable to that of MobileNetV2 on the VWW corpora [8] (within <1% variation) which uses 32-bit floats.

While it is possible to achieve similar inference performance of TinyML models (32-bit float conversion to 8-bit fixed-point) through pre- or post processing approaches, reducing the bit-precision to fewer than 8 bits degrades the performance significantly [26, 29]. To mitigate this problem, researchers have explored mixed-precision fixed-point numerical format [10, 11, 14, 17, 30, 31, 37]. However, utilizing mixed-precision fixed-point required a precision-assignment policies for TinyML model parameters across layers. For instance, Rusci *et al.* leveraged the combination of reinforcement learning and QAT to automatically select the appropriate precision of TinyML parameters across layers [30]. And thus demonstrates that it is possible to evaluate even the ImageNet corpora using mixed-precision fixed-point formats on MobileNetV2, with results within <1% variation as compared to inference with 32-bit floats.

This research proposes a tapered fixed-point quantization algorithm for TinyML models where the dynamic range of this numerical format is adapted to the dynamic range of the TinyML model parameters. A notable difference between this work and previous works is that this quantization approach does not required complex hardware or algorithmic approaches such as precision-assignment policies, QAT or calibration.

## 4 TENT FRAMEWORK

The goal of the TENT empirical framework is to emulate ML inference with quantization using low-precision tapered fixed-point format. Formally, the TinyML model parameters are quantized to tapered fixed-point such that the dynamic range and distribution of TinyML model parameters matches the distribution of values represented by tapered fixed-point. Therefore, the TENT framework, as shown in Figure 2, approximates the optimal tapered fixed-point numerical format for TinyML model parameters in each layer by selecting appropriate $IS$ and $SC$ values. These parameters are selected based on dynamic range of TinyML model parameters in each layer. After this step, the learned weights and activations (32-bit floats) are quantized to the low-precision tapered fixed-point format, upon which the dot product operations are then carried out.

In particular, the TENT empirical framework comprises of three key aspects: Tapered fixed-point parameters selection, quantization to tapered fixed-point, and the low-precision tapered fixed-point dot product.

### 4.1 Tapered Fixed-Point Parameter Selection

Algorithm 1 presents the $IS$ and $SC$ optimization procedure. To select these parameters, in the first step, the maximum absolute value of the DNN parameters at each layer ($W_{amax}$, and $A_{amax}$) are computed (lines 2-3) and rounded up to generate the appropriate $IS$ value. Algorithm 1, defines the process by which $IS$ is selected to tailor the numerical format to the range and distribution of parameters in each individual layer. In the worst case scenario, when the dynamic range of DNN parameters is larger than dynamic range of tapered fixed-point, the maximum possible value $n$ (total number of bits represented in tapered fixed-point format) is selected for $IS$

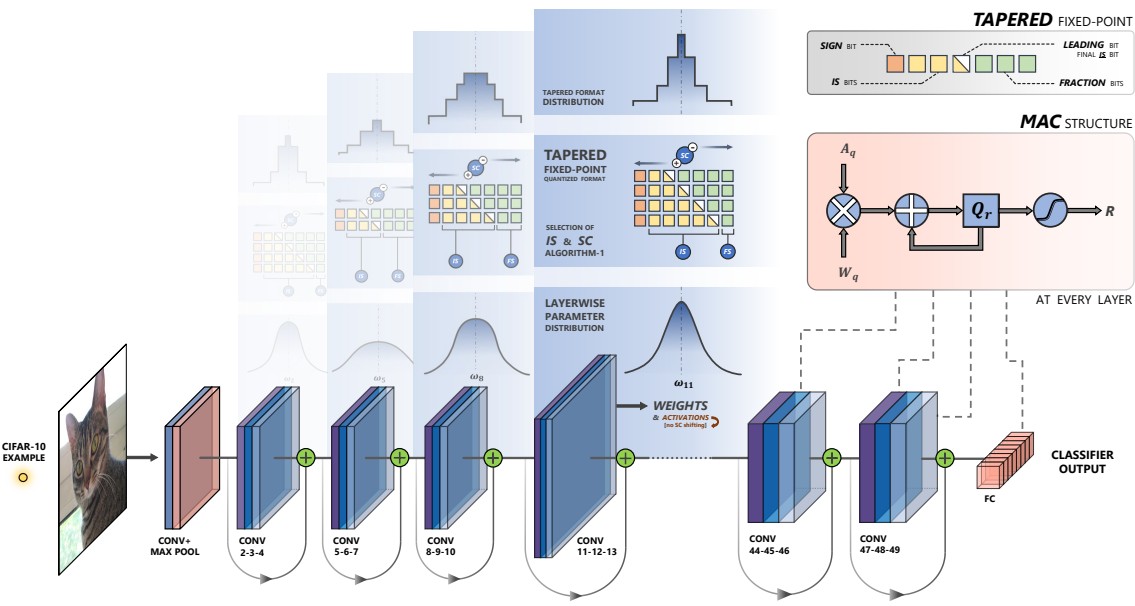

**Figure 2: TENT: a low-precision framework for TinyML inference with tapered fixed-point parameters. The framework is applied to each layer individually, selecting specific IS and SC values to match the distribution and range of parameters within the layer. IS specifies the *maximum* number of integer bits, and SC specifies the degree of shift required (left-shift if positive, right-shift if negative). The MAC structure displays the multiply-and-accumulate unit explained in figure 3.**

(lines 4-13). Selecting *IS* based on the algorithm 1 reduces the over-flow error in quantization. However, when the maximum absolute value of a DNN parameter is less than the maximum absolute value representable by the tapered fixed-point format selected; many bit-patterns in the numerical format are unused. To mitigate this issue, the the maximum absolute value of tapered fixed-point format is scaled down by 2 raised power of *SC* that determines as a base-2 logarithm of $w_{amax}$ (lines 14-21).

## 4.2 Quantization with Tapered Fixed-Point

In this paper, the quantization function $Q(x_i, q)$ defined in (4) approximates each parameter $x_i$ to $x_i'$ (a q-bit tapered fixed-point). In the quantization procedure, the values that lie outside the dynamic range of a given tapered fixed point format configuration, $Q(\cdot, \cdot, \cdot, \cdot)$, clips to the format maximum ($u$) or minimum ($l$) appropriately. A value that is between consecutive tapered fixed-point numbers is rounded to the nearest even number ($RNE(x_i)$).

$$x_i' = Q(x_i, l, u, q) = \begin{cases} l, & x > l \\ RNE(x_i), & l > x > u \\ u, & u > x \end{cases} \quad (4)$$

## 4.3 Tapered Fixed-Point Dot Product

The tapered fixed-point dot product is presented in Algorithm 2. In the first step, a set of quantized weights and activations are decoded to the tapered fixed-point (lines 2,3). To decode the tapered fixed point format, the sign bit, integer bits (through leading zero detection algorithm), and remaining fractional bits require to be extracted. Then, the product of the tapered fixed-point weights and activations are calculated without truncation or rounding after

---

**Algorithm 1** Compute the maximum integer bit width (*IS*) and scaling factor (*SC*) of tapered fixed-point for DNN parameters
**Input**: layers weights ($W_l$), layers activations ($A_l$)
**Output**: $IS_{W_l}, IS_{A_l}, SC_{W_l}$ tapered fixed-point parameters

---

1: **procedure** *IS*, *SC* SELECTION ($W_l, A_l$)
2:      $w_{amax} \leftarrow \max(|w_l|)$
3:      $A_{amax} \leftarrow \max(|A_l|)$

---

     Compute the $IS_{W_l}, IS_{A_l}$
4:      **if** $\lfloor W_{amax} \rfloor + 1 < n_{bit}$ **then**
5:          $IS_W \leftarrow \lfloor W_{amax} \rfloor + 1$
6:      **else**
7:          $IS_W \leftarrow n_{bit}$
8:      **end if**
9:      **if** $\lfloor A_{amax} \rfloor + 1 < n_{bit}$ **then**
10:        $IS_A \leftarrow \lfloor A_{amax} \rfloor + 1$
11:      **else**
12:        $IS_A \leftarrow n_{bit}$
13:      **end if**

---

     Compute the $SC_{W_l}$
14:      $SC_{W_l} \leftarrow 0$
15:      **if** $W_{amax} < 1$ **then**
16:        $SC_W \leftarrow |\lfloor \log_2(W_{amax}) \rfloor| + 1$
17:      **end if**
18:      **return** $IS_{W_l}, IS_{A_l}, SC_{W_l}$
19: **end procedure**

---

multiplication operations (lines 6-11). The products are then stored

**Algorithm 2** Tapered fixed-point dot product operations for $n$-bit inputs each with $\log n$ RS bits, $\log n$ SC bits.

**Input**: layers quantized weights ($W_{l_q}$), layers quantized activations ($A_{l_q}$),

**Output**: $R$ as Dot Product result

1: **procedure** TAPERED FIXED-POINT DP ($W_{l_q}, A_{l_q}$)
2:     $\text{sign}_w, \text{Int}_w, \text{frac}_w \leftarrow \text{DECODE}(W_{l_q}, IS_w)$
3:     $\text{sign}_a, \text{Int}_a, \text{frac}_a \leftarrow \text{DECODE}(A_{l_q}, IS_a)$

    **Multiplication**
4:     $\text{sign}_{mult} \leftarrow \text{sign}_w \oplus \text{sign}_a$
5:     $\text{Value}_w \leftarrow (\text{Int}_w + \text{frac}_w) \ll \text{frac}_{bit_w} + SC_w$
6:     $\text{Value}_a \leftarrow (\text{Int}_a + \text{frac}_a) \ll \text{frac}_{bit_a}$
7:     $p_{mult} \leftarrow \{\text{sign}, \text{Value}_w \times \text{Value}_a\}$

    **Accumulation & Normalize**
8:     $\text{sum}_{quire} \leftarrow p_{mul} + \text{sum}_{quire}$     ▷ Accumulate
9:     $\text{sum}_{nquire} \leftarrow \text{sum}_{quire} \gg \text{frac}_a + \text{frac}_w + SC_w$

    **Rounding & Encode**
10:    $\text{result} \leftarrow \text{ROUNDING \& ENCODING}(\text{sum}_{nquire})$
11:    **return** result
12: **end procedure**

in a wide register (*quire*[13]) for $m$ multipliers with size of $w_{quire}$ as in (5) (lines 12-15).

$$w_{quire} = \lceil \log_2(m) \rceil + 2 \times \lceil \log_2(\frac{Max_{TFX}}{Min_{TFX}}) \rceil + 2 \qquad (5)$$

The stored products are then converted and accumulated with fixed-point arithmetic. At the end, the accumulated result is encoded back into the tapered fixed-point numerical format (lines 16-18).

## 5 SYSTEM DESIGN AND ARCHITECTURE

The TENT framework gives insights into adopting a new numerical format for storing the weights and activations which reduce the quantization loss at low precision. This section describes the framework designed to simulate DNNs on hardware platforms in order to evaluate the performance of the tapered fixed-point representation in terms of latency and energy consumption. Figure 3 shows the high-level architecture of the designed framework guided by the design of Eyeriss v2 [6]. Primarily, it is composed of processing elements (PEs) arranged in a 2D systolic array architecture accompanied by a hierarchical memory organization. Systolic architectures have shown promising results in computing convolutions at low energy cost due to the data reuse characteristics and parallel processing features [19]. Most systolic architectures, commonly used to perform convolution operations, adopt input stationary, weight stationary, and output stationary dataflows. Of which, output stationay has shown to reduce execution time and energy consumption when PE computations are confined to a single pixel in the output feature map, and was thus selected for this architecture.

The PE in the systolic array has a tapered fixed-point based MAC unit with configurable bit-precision. It is controlled by an external control signal (*IS*) which defines the integer value and dynamic range of the format. Each MAC unit first reads an N-bit activation and weight in tapered fixed-point format and decodes them to $\lceil log_2(n) \rceil + n$ bits fixed-point value with $\lceil log_2(n) \rceil + 1$ integer bits and $n - 2$ fraction bits. The decoding process utilizes a count leading zeros (CLZ) unit, which counts the total number of leading zeros (from MSB), representing the integer value of the fixed-point as shown in 4. Furthermore, the 3-bit *SC* signal controls the scaling factor of the weight by shifting the fixed-point number based on the magnitude and the sign of the *SC* signal. The decoded activations and scaled weights are multiplied using a fixed-point multiplier which is further accumulated. Unlike other dataflows, the output stationary dataflow does not require the quantization of the accumulated value at every step which avoids generation of partial sums and thus eliminates the quantization error which normally occurs in intermediate stages. The accumulated fixed-point value uses a unary encoding mechanism, based on the magnitude and sign of the integer, while converting to the n-bit tapered fixed-point format.

The memory hierarchy consists of 128 MB off-chip main memory (DRAM) and 3×108 kB on-chip scratchpad memory (SRAM). The main memory is dedicated to storing input data, activations and weights/filters that are loaded by the host processor, whereas the scratchpad memory serves as a global buffer. In order to estimate latency, we bridge our framework with the SCALE-Sim tool [32]. SCALE-Sim, however, does not consider the cycles consumed in shuttling data back and forth between the global buffer and the DRAM. Therefore, the total latency is re-approximated by considering PE array execution time and DRAM access time (Micron MT41J256M4). For energy estimation analysis, calculation of execution time, and power utilization, we factor in 45nm CMOS technology node.

## 6 EXPERIMENTAL SETUP, RESULTS & ANALYSIS

The TENT framework is implemented in C++ and extended to support the TensorFlow framework [1]. To demonstrate the efficacy of the TENT framework, the performance of low-precision tapered fixed-point is evaluated on three inference tasks and is compared to the low-precision fixed-point numerical format. The specifics of the evaluation tasks and the inference performance achieved on them with 32-bit float DNNs are summarized in Table 1. The MNIST and CIFAR-10 datasets are chosen for evaluation, as they are ideal for tinyML applications [3]. To appropriately evaluate the benefits of the tapered fixed point format, the Fashion-MNIST dataset is also considered which presents additional challenges to the classification task, while still falling under a similar category as the MNIST dataset. The base model selected to perform the classification task on these datasets is an extremely compact model containing less than 2 million parameters that can be stored on tiny edge devices (*eg.*, ARM M-7 MCU) [2]. In the evaluation of each format, $IS \in [1..n-1]$ and $SC \in [0..n-1]$ are considered for tapered fixed-point, and $I \in [1..n-1]$ and $fs \in [1..n-1]$ are considered for standard fixed-point.

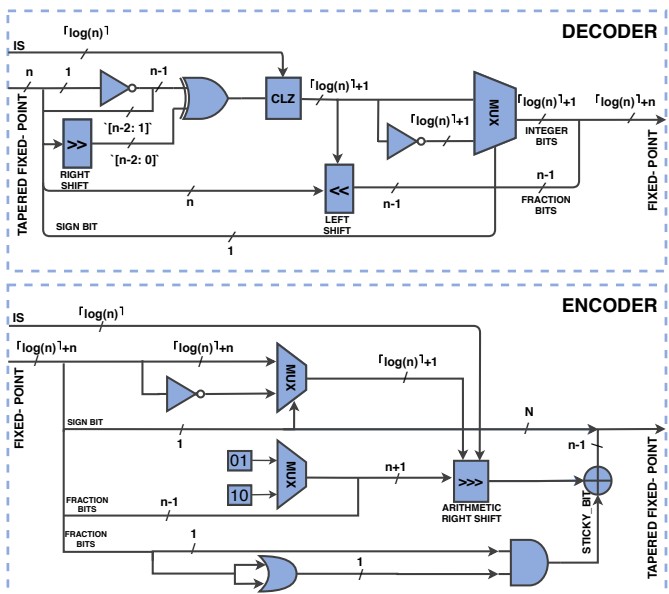

**Figure 3: Deep Neural Network accelerator architecture for TENT, with custom tapered fixed-point processing elements. The architecture is evaluated in a full cycle-emulator to analyze the performance and energy constraints.**

**Figure 4: RTL design for decoder, converting tapered point precision to fixed- point with count leading zero governed by *IS* parameter and encoder, converting fixed-point to tapered fixed-point with rounding off fraction bits to nearest value.**

## 6.1 Inference Performance with TENT

The efficacy of the TENT framework is evaluated on DNN inference with varied *IS* and *SC*, as shown in Table 2. The findings show that the low-precision tapered fixed-point outperforms the standard fixed-point on various benchmarks by up to ≈31%. For instance, the performance of an 8-bit low-precision tapered fixed-point ResNet-18 network on the CIFAR-10 dataset is improved by 27.44% compared to the fixed-point based network. Furthermore, we observed that the tapered fixed-point shows greater benefits on TinyML models whose parameters have a large dynamic range, such as ResNet model shown in Table 1 and Table 2. These performance benefits can be intuitively explained by the auto-tuning capability of the TENT framework, which adapts the format to the

dynamic range of the weights and activations, so as to reduce the quantization error. The best performance observed on all the benchmarks (when analyzed across the full [5..8]-bit range) is achieved with tapered fixed-point.

## 6.2 Hardware System Results

The execution time of the DNN model is mainly governed by the dataflow and the PE array architecture. The output stationary dataflow offers a 24% reduction in latency as compared to weight stationary dataflow while performing inference [32]. For a compute-bound DNN this is a significant improvement, considering that inference favors latency over throughput [28]. The homogeneous 16×16 PE configuration with no sophisticated architectural features offers

**Table 1: Description of the TinyML models and benchmarks using 32-bit float parameters.**

| Dataset | TinyML Model | W-Range [1] | A-Range [1] | # Parameters | # MACs [2] | Performance |
|---|---|---|---|---|---|---|
| MNIST | ConvNet [3] | $[-0.78, 0.62]$ | $[0, 3.61]$ | 1.40 M | 58.7 K | 99.32% |
| Fashion-MNIST | ConvNet [4] | $[-0.74, 0.45]$ | $[0, 5.97]$ | 1.88 M | 69.8 K | 92.54% |
| CIFAR-10 | ResNet-18 | $[-2.12, 1.17]$ | $[0, 10.21]$ | 0.27 M | 286.72 K | 91.54% |

[1] W: Weights; A: Activations
[2] The number of MACs are calculated for a DNN inference with a batch size of 1.
[3] 2 Convolutional layers, 2 fully-connected layers, and 1 Pooling layer
[4] 3 Convolutional layers, 2 fully-connected layers, 2 Pooling layers, 1 Batch normalization layer

**Table 2: Performance of TinyML models during inference with the tapered fixed-point(TFX) and fixed-point.**

| Bit Precision | MNIST | | Fashion-MNIST | | CIFAR-10 | |
|---|---|---|---|---|---|---|
| | TFX | Fixed-Point | TFX | Fixed-Point | TFX | Fixed-Point |
| 8-bit | 99.33% | 99.18% | 92.59% | 89.59% | 81.66% | 54.20% |
| 7-bit | 99.32% | 97.14% | 92.47% | 88.63% | 75.90% | 24.50% |
| 6-bit | 99.30% | 97.08% | 92.14% | 85.31% | 46.79% | 12.96% |
| 5-bit | 99.29% | 96.96% | 89.35% | 83.46% | 23.44% | 11.11% |

improvement in computing efficiency from 89.58% to 91.82%, with a significant reduction in energy consumption. Figure 5(c) illustrates the energy-delay product (EDP) for the tapered fixed-point network ResNet-18 while performing inference on the CIFAR-10 dataset. It is worth noting that tapered fixed-point offers 31% improvement in classification accuracy with a negligible EDP overhead ($17 - 30\%$) as compared to fixed-point. Reduced bit-precision economizes the local memory storage size and the number of operational cycles in both tapered and standard fixed-point.

Overall, the use of the tapered fixed-point numerical format helps in paving a path towards minimizing the power consumption (inference), which aligns with the ultimate goal envisioned by the TinyML community ($\leq 1$ mw [3]). Our analyses have shown that this work's adaptation of the ResNet-18 architecture consumes power within the range of 189-270 mw to classify a single sample of the CIFAR-10 dataset, which befits the low-power requirements of devices that operate on the edge [2]. Additionally, the proposed TENT framework further enhances the performances by leveraging the flexibility of a tapered precision numerical format and its affinity to represent parameters with reduced quantization error (as compared to standard fixed point formats).

# 7 CONCLUSIONS

This paper presents a low-precision framework, TENT, that offers quantization using tapered fixed point to perform inference on TinyML models. The maximum integer bit width and scaling factor parameters are dynamically selected to best fit the variability of the parameter and activation distributions within each layer of the model. We observe reduction in total quantization error that leads to an improvement in inference accuracy by $\approx 31\%$ over fixed-point models. Furthermore, we show that tapered fixed-point achieves

this with a moderate increase in energy consumption over fixed-point.

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
