# OpenReview forum: "TENT: Efficient Quantization of Neural Networks on the tiny Edge with Tapered FixEd PoiNT"
_tinyml.org/tinyML/2021/Research_Symposium — tinyML 2021 Poster_

### Official Review · AnonReviewer1 · 2021-01-28

**Overall Merit Score:** 2

**Brief Summary:**

* Proposal of a new number format: Tent, which is a tapered fixed point number format (with hence a tent-shaped precision number). Claims to be better then fixed point and posit. (drawing inspiration from both).
* Also proposing a hardware architecture (mostly encoder and decoder) for this.
* Experiments with MNIST, FashionMNIST and CIFAR10 (with ResNet18).

**Detailed Comments:**

* Paper explanations are absolutely not clear. It is very hard to decipher what has exactly been done and what claims are made for what reasons.  Also the new number format is not clearly explained (what is supposed to be the core of the paper):
    - Section w talks about the TENT format, but does not mention nor explain SC. This comes into the picture all of a sudden later in the paper, at the point where it is mentioned that SC has to be optimized...
    - One claimed reason to use this data format, is that it would not need precision assignment policies, yet IS and SC are determined layer-wise, which means that conversions between layers are still needed. This is similar to dynamic fixed point.
    - I lack to see the benefits vs dynamic fixed point.
    - How does a layer knows the needed SC and IS for its outputs?
* The simulation framework seems to make some non HW-feasible abstractions: E.g.
    - Round to nearest even number is very difficult in HW. Why not floor, or round to zero?
    - Is matching between HW simulations and the ML framework verified?
* Experiments not clearly explained and experiments missing:
    - As the paper claims to be a better alternative compared to posit, a comparison with posit should have been done
    - It is not clear how the competitive fixed point simulations are set up. What quantization rule is used here? Is it dynamic fixed point? Are the scales chosen layer wise, or channel-wise, or ...? It this done with post training quantization, or quantization aware training?  Etc.?
    - The TENT data format needs an IS and SC parameters, which is not counter in the number of bits for a parameter. Is such a scaling parameter also added to the dynamic fixed point comparison point?
    - why are IS, SC, I and fs all using the same datalength (n bits? pg 4 bottom right)
* The described hardware for encoder/decoder (fig 4) is difficult to understand. The pictures (and the text) also seems to mix "n" and "N", which is adding to the confusion (are they different symbols: n and N?)

**Paper Strengths:**

* Encompasses number format, training experiments and hardware considerations.

**Paper Weaknesses:**

* Paper explanations are absolutely not clear. It is very hard to decipher what has exactly been done and what claims are made for what reasons.  Also the new number format is not clearly explained (what is supposed to be the core of the paper):
* The simulation framework seems to make some non HW-feasible abstractions
* Experiments not clearly explained and experiments missing:
* The described hardware for encoder/decoder (fig 4) is difficult to understand. The pictures (and the text) also seems to mix "n" and "N", which is adding to the confusion (are they different symbols: n and N?)

**Poster (If Paper Is Rejected):**

1: Yes, ok for poster sesion to nurture work

**Reviewer Confidence:**

4: The reviewer is confident but not absolutely certain that the evaluation is correct

---

### Official Review · AnonReviewer4 · 2021-01-29

**Overall Merit Score:** 1

**Brief Summary:**

The paper proposes a modified Posit number format for realizing CNNs in hardware. Specifically, the regime bits in Posit is replaced with signed unary encoding. The authors claim that using such format for ResNet-18, classification accuracy improves by ~31% as compared to fixed-point with ~17-30% energy overhead.

**Detailed Comments:**

To summarise, the paper should pick an appropriate baseline for comparison and study the proposed number format. Furthermore, since this is an improvement over standard posit based implementation, the study must show a detailed comparison against more established posit-format.

The paper does not describe the generic properties on the newly proposed number format in enough detail to be useful for the community. It should use some of the circular number line formats to show the pros and cons of the proposed number format and draw a contrast with posit.

See additional comments above.


**Paper Strengths:**

The basic reasoning on why the authors came up with the TENT number format is interesting.

**Paper Weaknesses:**

The paper compares accuracy with equivalent bit-width fixed-point number format. This comparison is misleading because the drastic loss in Top-1 accuracy is unacceptable though it is better than other poorly implemented systems. Crucially, the paper should compare the accuracies of TENT-based low-bit-width implementation with widely available FP32 baseline.

For CIFAR10 dataset, sub-7-bit implementation is practically not useful as accuracy is extremely poor. In such scenarios, neither the fixed-point nor the TENT based system provide any meaningful accuracy that could be useful for real-life applications.

The paper does not discuss how "rounding" is implemented and its effect on the overall accumulation of dot products. This is probably the most important aspect of any new number format arithmetic. How is "double rounding" avoided in TENT-based implementation?

The accumulator design is very minimalistic and does not contain details. For example: How is the long chain of carries propagated, how does the accumulator's working frequency depend on the total number of bits?

Figure 5 compares the area between a fixed-point based MAC unit and Tapered fixed-point based MAC unit. This comparison does not make sense as accuracy achieved in each case is very different. The paper should rather compare the cost of MAC unit of two equally accurate (classification accuracy) implementations (e.g. FP32/16 system and tapered fixed-point based system).

The paper claims that the TENT number format's dynamic range and distribution matches with the distribution of the DNN model parameters. This claim should be proven through empirical study. The authors should make an ablation study to show how many bits should be dedicated to fraction to match the real DNN model parameter distribution.


**Poster (If Paper Is Rejected):**

1: Yes, ok for poster sesion to nurture work

**Reviewer Confidence:**

4: The reviewer is confident but not absolutely certain that the evaluation is correct

---

### Official Review · AnonReviewer3 · 2021-01-29

**Overall Merit Score:** 2

**Brief Summary:**

In this paper, the authors proposed TENT, a framework that quantizes deep learning models with tapered fixed-point, which achieves better accuracy at the same bit-precisions compared to fixed-point quantization. They also proposed a custom hardware architecture to study the latency and energy trade-off.

**Detailed Comments:**

The paper proves that the TFX data format can improve the accuracy compared to fixed-point under some quantization method. However, the paper lacks solid comparison to baselines using custom data format, and baselines on quantization algorithm+ custom hardware solution. Without a solid comparison, it is hard to demonstrate the value of the proposed method. Therefore, I lean on rejection for now.

**Paper Strengths:**

1. Firstly, the paper is generally well written and easy to read.
2. It provides not only a new quantization method but also a hardware accelerator to support the inference. A comprehensive hardware statistics comparison is provided.


**Paper Weaknesses:**

1. The major novelty of the proposed method is to use a TFX data format instead of normal fixed-point. However, the TFX format needs specialized hardware for acceleration, unlike fixed-point that can be supported with existing hardware like GPUs/CPUs/ARM cores. Given specialized hardware is needed, the author should discuss the comparison to existing works that also use a custom data format for quantization (e.g., codebook-based quantization in [a]) and show the possible advantage of TFX format.
2. The fixed-point baseline used in this paper seems too weak. For example, the 8-bit fixed-point CIFAR-10 results in Table 2 suffer an accuracy drop of 37.3%, which is not very reasonable. For 8-bit quantization, the accuracy drop is usually negligible [16]. The authors should use stronger baselines like quantization-aware training with PACT [7] (or [16] if post-training quantization).
3. The paper only includes small and toy datasets, which fails to reflect the real-life use cases. Even for TinyML setting, there are more realistic datasets like Visual Wake Words and Google Speech Commands.
4. The proposed work needs a co-design of the algorithm and the hardware. Therefore, the authors should compare to other algorithm+hardware solutions to compare the overall accuracy v.s. energy/area/... trade-off, like [b, c, d]

[a] Han et al., Deep Compression: Compressing Deep Neural Networks with Pruning, Trained Quantization and Huffman Coding

[b] Park et al., Energy-Efficient Neural Network Accelerator Based on Outlier-Aware Low-Precision Computation

[c] Sharma et al., Bit Fusion: Bit-Level Dynamically Composable Architecture for Accelerating Deep Neural Networks

[d] Sharify et al., Laconic deep learning inference acceleration

**Poster (If Paper Is Rejected):**

1: Yes, ok for poster sesion to nurture work

**Reviewer Confidence:**

5: The reviewer is absolutely certain that the evaluation is correct and very familiar with the relevant literature

---

### Official Review · AnonReviewer2 · 2021-01-30

**Overall Merit Score:** 2

**Brief Summary:**

This paper proposes to use a tapered fixed point numerical representation for deep neural networks (DNNs) where the numerical format's dynamic range and distribution match DNN's parameter distribution. The paper proposes a quantization algorithm to generate this representation and also modifies an existing hardware architecture to implement tapered fixed point multiply accumulates, and obtains system level accuracy and performance metrics in simulation.

**Detailed Comments:**

- This paper proposes to use a tapered fixed point numerical representation for deep neural networks (DNNs) where the numerical format's dynamic range and distribution match DNN's parameter distribution. The paper proposes a quantization algorithm to generate this representation and also modifies an existing hardware architecture to implement tapered fixed point multiply accumulates, and obtains system level accuracy and performance metrics in simulation.

- The concept of using tapered fixed point for DNNs is interesting, and the paper proposes a full stack framework from model quantization all the way to hardware evaluation.

- However, the claims the paper makes about accuracy improvement over fixed point are exaggerated. It is unclear how the paper performs the fixed point quantization in Table 2 for comparison purposes. 54.2% accuracy on CIFAR-10 with 8 bit fixed point ResNet-18 is far below the state of the art. Existing fixed point quantization methods achieve close to 90% accuracy, which is 8% higher accuracy than TENT and has a lower hardware cost.

- More complex quantization methods such as LQ-Nets (https://openaccess.thecvf.com/content_ECCV_2018/papers/Dongqing_Zhang_Optimized_Quantization_for_ECCV_2018_paper.pdf) from 2018 achieve 90.2% accuracy with only 2 bit weights and 2 bit activations on ResNet-20 for CIFAR-10.

- The tapered fixed-point format is not explained clearly in Section 2. A lot of the symbols used in the notation are undefined (x, D_TFX, Is, fs, I is used to represent the bit as well as the value of the integer part in equation 3). SC symbol is mentioned but its effect is not taken into account in equation 2. The reader is required to read a portion of reference [13] to understand this format. This section can be greatly improved to enhance readability.


**Paper Strengths:**

- The concept of using tapered fixed point for DNNs is interesting.

- The paper proposes full stack framework from model quantization all the way to hardware evaluation.

**Paper Weaknesses:**

- Comparison baseline is incorrect and far from the state of the art.

- There is not much novelty in the hardware architecture.

**Poster (If Paper Is Rejected):**

1: Yes, ok for poster sesion to nurture work

**Reviewer Confidence:**

5: The reviewer is absolutely certain that the evaluation is correct and very familiar with the relevant literature

---

### Decision · Program_Chairs · 2021-02-05

**Decision:**

Accept (Poster)

**Comment:**

Based on the reviewer feedback, your paper has been accepted as a poster.

Please read the reviews carefully and make sure the concerns are addressed in your poster submission.

Accepted posters are given a 5-minute slot for an oral presentation on Friday, March 26, 2021, to pitch the main ideas of your work and to stimulate discussions. Detailed instructions will follow soon. All final posters will earn a stamp of acceptance as such: “Published as a poster at TinyML Research Symposium 2021.”